# Constitutive *OsCIN1* Expression Reprograms Source–Sink Dynamics and Compromises Agronomic Traits in Rice

**DOI:** 10.3390/ijms262311471

**Published:** 2025-11-27

**Authors:** Cong Danh Nguyen, Joon-Seob Eom, Jung-Il Cho, Seok-Hyun Choi, Jae Ung Kwak, Seong-Cheol Eom, Kieu Anh Thi Phan, Juho Lee, Jong-Seong Jeon, Sang-Kyu Lee

**Affiliations:** 1Division of Life Science, Plant Molecular Biology and Biotechnology Research Center, Gyeongsang National University, Jinju 52828, Republic of Korea; ndmonkeyluffy@gmail.com (C.D.N.); contamkwak@gnu.ac.kr (J.U.K.); esc2455@gmail.com (S.-C.E.); phanthikieuanh95@gmail.com (K.A.T.P.); dlwngh1691@gnu.ac.kr (J.L.); 2Graduate School of Green-Bio Science and Crop Biotech Institute, Kyung Hee University, Yongin 17104, Republic of Korea; june031100@gmail.com (J.-S.E.); jungilcho@korea.kr (J.-I.C.); lilchoi92@gmail.com (S.-H.C.)

**Keywords:** carbon partitioning, cell wall invertase, *OsCIN1*, rice, sink strength, yield

## Abstract

Cell wall invertases (CINs) establish sucrose gradients between source and sink tissues, essential for the allocation of photoassimilates. Rice possesses nine *CIN* genes, among which *OsCIN1* and *OsCIN2* have been reported as key regulators of sink strength. To test whether increasing CIN activity enhances grain yield, we generated *OsCIN1* overexpression lines in rice driven by the CaMV 35S promoter. Subcellular localization analysis of *OsCIN1*–GFP confirmed its apoplastic localization. *OsCIN1* promoter::GUS analyses verified expression in vascular tissues and revealed predominant signals in the ovular vascular and lateral stylar vascular traces during seed development. Although CIN activity was markedly elevated throughout the plant, the resulting phenotypes were unexpected. Sugar profiling of flag leaves at the flowering stage showed almost complete sucrose depletion in the overexpression (OX) lines, accompanied by increased hexose and starch accumulation. Under field conditions, *OsCIN1* OX plants exhibited ~50% fewer tillers and a lower 1000-grain weight relative to wild type (WT), resulting in reduced productivity. Ectopic expression of *OsCIN1* disrupted the sucrose concentration gradient, weakened carbon partitioning to sink tissues, and impaired key agronomic traits. Collectively, sugar flux is governed by the spatiotemporal patterning of CINs, highlighting that precise spatial and temporal control of CIN activity is required to increase yield.

## 1. Introduction

Photosynthesis is the fundamental process that produces carbohydrates essential for plant growth and development [1,2]. The carbon assimilates synthesized in mesophyll cells are either temporarily stored in chloroplasts as starch for catabolic use or converted into sucrose in the cytosol for long-distance transport [3,4]. The sucrose thus produced is translocated through the phloem to sink organs such as roots, fruits, and grains, where it serves as a critical determinant of plant growth and final yield [5]. Therefore, not only the amount of carbohydrates produced by photosynthesis but also the efficiency of their partitioning from the source leaves to sink tissues such as grains has long been recognized as a key target for enhancing crop yield [6,7].

Long-distance sucrose transport is regulated by two major mechanisms: transporters and concentration gradients. Among these, sucrose transporters (SUTs) and Sugars Will Eventually be Exported Transporters (SWEETs) play particularly critical roles [8,9,10,11]. These transporters facilitate phloem loading by actively or passively mediating sucrose transport in bundle sheath cells and elements of the leaf vasculature [8]. For example, in the C4 photosynthetic system of maize (*Zea mays* L.), *ZmSWEET13* effectively exports sucrose from bundle sheath cells into the phloem [12], whereas in wheat (*Triticum aestivum* L.), TaSUT1 plays an essential role in the remobilization of stored carbohydrates [13]. The regulatory control of these transporters determines the flux of sucrose, which in turn represents a critical molecular mechanism governing crop yield [14]. In fact, a study in which the *Arabidopsis AtSUC2* gene was overexpressed in rice reported an increase in phloem loading efficiency, resulting in up to a 16% improvement in yield, thereby providing direct evidence that enhancing sucrose loading can substantially increase crop productivity [6].

For the efficient transport of sucrose, it is essential to maintain a concentration gradient between source and sink tissues. The enzyme that plays a pivotal role in establishing this gradient is the cell wall invertase (CIN). CINs are predominantly localized in the apoplastic space of sink organs such as developing seeds, where they rapidly hydrolyze sucrose imported through the phloem into glucose and fructose [15,16,17,18]. Because CIN is the only enzyme that hydrolyzes sucrose in the apoplast, it plays a central role in maintaining the concentration gradient and is considered the most critical enzyme that maintains the apoplastic sucrose gradient immediately before import into sink organs [19,20,21,22].

Given the pivotal role of CIN in regulating carbon allocation, numerous attempts have been made to enhance its activity to promote sink organ development [18,23]. In tomato, overexpression of *CIN* driven by a fruit-specific promoter resulted in increased fruit weight and overall yield [13]. Furthermore, disruption of a cell wall invertase inhibitor (INVINH) led to elevated sugar content without reducing weight in tomato [21,24]. In potato, suppression of *INVINH* expression was shown to increase the size of microtubers [25]. In carrot, reduced *CIN* expression impaired early developmental processes [26]. In maize, overexpression of *CIN* genes from *Arabidopsis*, rice, and maize consistently resulted in increased seed size [27]. In rice, expression of *OsCIN2* under the control of its native promoter significantly enhanced seed size [20]. Collectively, these studies demonstrate that manipulation of CIN activity or its regulatory components has profound effects on sink organ development and final yield across diverse plant species.

Recently, we identified the function of *OsCIN1.* The *oscin1/oscin2* double mutant exhibited a much more severe shrunken seed phenotype compared with the opaque phenotype of the *oscin2* single mutant, suggesting that *OsCIN1*, like *OsCIN2*, regulates carbon partitioning from maternal to filial tissues and is likely involved in governing sugar transport and partitioning to sink organs, particularly seeds [22]. Therefore, this study aimed to reprogram source–sink sucrose partitioning through overexpression of *OsCIN1* and to assess its impact on overall yield in rice.

## 2. Results

### 2.1. Generation of OsCIN1 Overexpression Lines

To analyze the function of overexpressed *OsCIN1* in rice, we cloned the full-length coding sequence (CDS) of the *OsCIN1* gene under the control of the 35S promoter (Figure 1a). Following transformation, regenerated plants were selected on hygromycin-containing medium. Multiple independent transgenic lines exhibiting differential *OsCIN1* expression were obtained. Among these, an intermediate expresser *OsCIN1* OX-1 and the highest expresser *OsCIN1* OX-3 were identified by semi-quantitative RT-PCR (Figure 1b) and subsequently confirmed by quantitative RT-PCR (qRT-PCR) to quantify transcript accumulation (Figure 1c). Enzymatic assays were performed to quantify CIN activity in the two independent lines. Across both vegetative and reproductive stages, CIN activity in leaves and stem segments of the OX lines was consistently and significantly higher than that of the wild type (WT) (Figure 1d,e).

### 2.2. pOsCIN1::GUS Expresses Mainly in Vascular Tissues During Seed Development

To determine where *OsCIN1* functions, we examined its expression pattern using a p*OsCIN1*-β-glucuronidase (GUS) reporter transgene (p*OsCIN1*::GUS). GUS activity was observed predominantly in leaf veins (Figure 2a) and in the vascular tissues of the lemma and palea (Figure 2b). Strong expression was also observed in the ovular vascular trace and lateral stylar vascular traces during early grain-filling stages (Figure 2e). At 15 days after pollination (DAP), GUS activity was specifically localized to the ovular and lateral stylar vascular traces (Figure 2d). Carbon partitioning and sucrose metabolism are recognized as central processes for grain filling in crops. During seed development, vascular traces act as conduits that deliver essential resources, including water, mineral nutrients, and sugars from the parent plant to the developing embryo and endosperm. In rice, sucrose produced in photosynthetic leaves is transported through the phloem to the developing grain, where it enters the endosperm via the dorsal vascular bundle. The sucrose then moves along the nucellar epidermis and is transferred into the endosperm and embryo by sugar transporters, after which it is converted to storage starch. This coordinated sequence of transport and conversion is essential for grain filling and determines both grain weight and overall yield. These results indicate that *OsCIN1* plays an important role during the grain filling stage and seed development.

We examined GUS activity not only during the reproductive stage but also in young seedlings. Strong GUS activity was detected in the vascular bundles of the coleoptile and the roots, whereas in leaves the overall activity was weaker but still detectable in the vascular bundles (Appendix A). These results indicate that *OsCIN1* is expressed even in young seedlings and that its expression is preferentially localized to vascular tissues.

### 2.3. OsCIN1 Is Located in the Apoplast

Multiple studies have established that CINs are in the apoplast, where they hydrolyze sucrose into glucose and fructose [28,29]. They interact with cell wall matrix polysaccharides through noncovalent forces such as ionic and hydrogen bonds rather than being covalently anchored [30]. This activity shapes apoplastic sugar levels and maintains the sucrose gradient that governs source-sink carbon flux [19]. To determine whether *OsCIN1*, like other CINs, is localized in the apoplast, GFP-tagged *OsCIN1* (*OsCIN1*-GFP) was expressed in transgenic callus tissue and examined by confocal fluorescence microscopy. The GFP signal confirmed that *OsCIN1* localizes to the apoplast (Figure 3).

### 2.4. OsCIN1 Overexpression Alters Sugar Composition

Given that OsCIN catalyzes sucrose hydrolysis, we hypothesized that sugar composition would be altered in the *OsCIN1* OX lines. Sucrose, the primary product of photosynthesis, is synthesized in leaves during the day, transported through the phloem, and subsequently unloaded into sink organs such as roots, young panicles, and developing grains [31]. Because a significant proportion of the carbohydrates required for grain development in rice is supplied by the flag leaf [32], we quantified soluble sugars and starch in the flag leaves of *OsCIN1* OX lines at the flowering stage. Glucose, fructose, and starch levels were significantly higher in the *OsCIN1* OX lines than in the WT (Figure 4a,b,d). Our experimental design does not allow compartment-specific attribution of the increased hexoses to either the cytosol or the apoplast. However, given that hexose concentrations generally exhibit much smaller diurnal fluctuations compared with sucrose and remain relatively stable [33] and that sucrose was undetectable in the OX lines whereas it was present at appreciable levels in WT flag leaves (Figure 4c), the observed increase in hexoses is most plausibly attributable to enhanced sucrose hydrolysis in the apoplast in *OsCIN1* OX lines. The absence of detectable sucrose in the OX lines is likely attributable to sustained sucrose hydrolysis in the apoplast. Driven by concentration gradients, sucrose from the vacuolar pool effluxes into the cytosol and is subsequently exported across the plasma membrane into the apoplast, where it is continuously cleaved into hexoses. This sustained hydrolysis likely accelerates the depletion of the sucrose pool.

### 2.5. Overexpression of OsCIN1 Suppresses Plant Growth and Grain Yield in Rice Paddy

To assess the impact of *OsCIN1* overexpression on plant growth under field conditions, transgenic lines were cultivated in paddy fields, and their developmental characteristics were monitored. *OsCIN1* OX lines exhibited significantly stunted growth compared with WT plants (Figure 5a). Tiller number in the OX lines was approximately 50% lower than that of WT plants (Figure 5a,b). Both 1000-grain weight and grain size were significantly reduced (Figure 5c–g). The depletion of sucrose in both the apoplast and cytosol of source tissues suggests that little or no sucrose was available for translocation to sink organs. Consequently, constitutively high CIN activity in both source and sink tissues likely impaired the establishment of the sucrose concentration gradient required for efficient sucrose transport. This perturbation is expected to have resulted in reductions in tiller number, seed size, and grain weight.

## 3. Discussion

### 3.1. OsCIN1 Functions as a Canonical Cell Wall Invertase with Tissue-Specific Expression

Understanding the activity and regulation of CINs is fundamental to elucidating how plants establish and maintain sucrose concentration gradients between source and sink tissues. These gradients, characterized by high sucrose levels in photosynthetic source tissues and low sucrose concentrations in metabolically active sinks, are generated and sustained through elevated CIN enzymatic activity in sink organs. This knowledge is crucial for developing strategies to enhance sink strength and optimize resource allocation in crop plants.

Subcellular localization analysis using GFP fusion proteins confirmed that *OsCIN1* localizes to the apoplast (Figure 3). p*OSCIN1*::GUS line revealed that *OsCIN1* is expressed in vascular tissues, with particularly expressed in the ovular vascular trace and lateral stylar vascular traces during seed development (Figure 2). This vascular-specific expression pattern establishes *OsCIN1* as a canonical cell wall invertase that strategically regulates sucrose loading and unloading pathways. The strong expression in developing seed vascular tissues suggests a central role in controlling carbon flux into the grain, which determines rice yield. In addition, our recent functional genetic analysis demonstrated that the *oscin1/oscin2* double mutant exhibits severe reductions in both seed starch accumulation and pollen starch synthesis. Enzyme activity assays further showed that OsCIN activity in mature anthers was approximately 500-fold higher than in leaves [22]. Taken together, these findings indicate that *OsCIN1* and *OsCIN2* act as the primary regulators of sink strength during reproductive development. This functional importance, combined with the role of CINs in enhancing crop productivity across multiple species [18,23,27], led us to hypothesize that constitutively overexpression of *OsCIN1* would enhance sink capacity in developing seeds by maintaining high apoplastic sucrose hydrolysis activity, thereby accelerating carbon flux into grains and improving crop yield. To test this hypothesis, we generated transgenic *OsCIN1-OX* lines under the CaMV 35S promoter.

### 3.2. Constitutive OsCIN1 Overexpression Paradoxically Impairs Agronomic Traits

To examine whether the increased CIN activity could enhance productivity, we generated transgenic rice lines overexpressing *OsCIN1* under the control of the constitutive CaMV 35S promoter. The overexpression lines exhibited increased CIN activity in fully expanded leaves and stem segments at both vegetative and reproductive stages (Figure 1d,e). This confirmed that the 35S promoter effectively enhances *OsCIN1* expression and that the encoded enzyme retained full catalytic activity *in planta*. However, the phenotypic outcomes differed from our predictions. Rather than enhancing productivity, constitutive overexpression of *OsCIN1* resulted in significant agronomic penalties under field conditions. Tiller number, a key yield component in rice, was reduced by approximately 50% in OX lines compared to wild type (Figure 5a,b). Furthermore, 1000-grain weight decreased significantly (Figure 5c), accompanied by reductions in grain width and thickness (Figure 5d–g). These results demonstrate that ectopic, constitutive elevation of CIN activity throughout the plant does not strengthen sink capacity as anticipated. Instead, it disrupts developmental tillering processes and seed development, ultimately diminishing overall productivity.

### 3.3. The Importance of Spatiotemporal Regulation of CIN Expression

The mechanistic basis for these adverse phenotypes is revealed by sugar profiling of flag leaves, where the primary photosynthetic organ supplying carbon to developing grains in rice [32]. At the flowering stage, *OsCIN1* OX lines exhibited depletion of sucrose in flag leaves, while WT plants maintained suitable sucrose levels (Figure 4c). Simultaneously, glucose, fructose, and starch accumulated significantly higher levels in *OsCIN1* OX flag leaves compared to wild type (Figure 4a,b,d). These metabolic perturbations indicate the impairment of source-sink carbon partitioning. In wild-type plants, sucrose synthesized in source leaves is exported to the phloem and transported to sink organs, where localized CIN activity in the apoplast establishes the concentration gradient driving continued sucrose flux. However, when CIN activity is elevated throughout the plant, sucrose is excessively hydrolyzed, leading to the collapse of the sucrose gradient between source and sink organs. The hexoses generated by CIN overexpression cannot substitute for sucrose in long-distance transport and therefore accumulate in source tissues.

The complete absence of detectable sucrose in *OsCIN1* OX flag leaves indicates that sustained apoplastic sucrose hydrolysis creates a concentration gradient that continuously draws sucrose from intracellular pools (cytosol and vacuole) into the apoplast, where it is immediately cleaved. This sustained depletion eliminates the sucrose available for phloem loading and long-distance transport to sink organs. Consequently, despite high CIN activity in sink tissues, insufficient sucrose delivery results in carbon starvation of developing grains and other sink organs, explaining the observed reductions in tillering, grain size, and overall yield.

Consistent with our findings, several studies have reported that constitutive CINs overexpression disrupts normal plant development and yield traits. In cassava, overexpression of *MeCWINV3* disrupted source-to-sink sugar partitioning, reduced sugar movement to storage roots, suppressed starch biosynthesis gene expression, and affected the yield [34]. Similarly, heterologous expression of yeast invertase in plant cell walls has consistently been shown to disrupt assimilate allocation, impair root development in tobacco [35,36]. These studies demonstrate that continuous expression of invertase activity perturbs normal carbon partitioning patterns and impairs plant development. Most instructive is the case of rice *OsCIN2*, which shares high functional similarity with *OsCIN1*. When *OsCIN2* was overexpressed under the constitutive 35S promoter, transgenic plants exhibited shrunken seeds remarkably similar to loss-of-function mutants [20]. In contrast, when the same gene was expressed under its native promoter, grain size increased significantly [20]. These findings clearly demonstrate that the spatiotemporal pattern of CIN expression is a key determinant of phenotypic outcomes.

### 3.4. Functional Specialization Within the CIN Gene Family

The *CIN* gene family is involved in many plant species. There are six members in *Arabidopsis thaliana* [37], eight in tomato (*Solanum lycopersicum*) [38], 12 in tobacco (*Nicotiana tabacum*) [39], and nine in rice [40]. The large number of genes in each species suggests functional diversity rather than genetic redundancy. Different tissues and developmental stages require different metabolic pathway activation and different sucrose concentrations. The multiplicity of *CIN* genes enables plants to adjust sucrose hydrolysis to meet tissue-specific metabolic requirements, developmental stages, and environmental conditions (light intensity, temperature, water, and nutrients). To improve the yield by manipulating CIN activity, a comprehensive understanding is required for the tissue-specific, temporal dynamics, and expression level. Moreover, the enzyme’s catalytic properties, substrate availability, and integration with downstream pathways such as hexose transport and metabolism are also assumed. In parallel, the discovery, mechanistic characterization, and application of CIN inhibitors that enable post-translational regulation via competitive inhibition of CIN activity are required. Such a multilayered, precision-targeted strategy should circumvent the adverse outcomes of indiscriminate constitutive overexpression and, by enabling refined control of source-to-sink sucrose gradients, deliver stable and reproducible improvements in crop yield.

## 4. Materials and Methods

### 4.1. Plant Material

The japonica rice line *Oryza sativa* cv. Dongjin was used throughout this study. All plant materials were freshly harvested and stored at −80 °C prior to experiments. Sterilized seeds were placed on ½ Murashige and Skoog (MS) agar medium for germination at 28 °C for 7 days before being transferred to a greenhouse for cultivation under a 14 h light/10 h dark photoperiod.

### 4.2. Plasmid Construction

For the *OsCIN1* overexpressing construct, the full-length coding sequence of *OsCIN1* 1191 base pairs (bp) was amplified by polymerase chain reaction (PCR) from rice (*O. sativa* L. ssp. japonica cv. Dongjin) cDNA using Phusion High-Fidelity DNA Polymerase (Thermo Scientific, Waltham, MA, USA) and cloned into the pCR-4-TOPO vector (Invitrogen, Waltham, MA, USA). Following sequencing verification, the *OsCIN1*-TOPO clone was subcloned into a Gateway entry vector and placed between the 35S promoter and Nos terminator or GFP-Nos terminator using the Multi-Round LR recombinase-mediated Gateway™ system (Invitrogen, Waltham, MA, USA), generating p35S::*OsCIN1* and p35S::*OsCIN1*-GFP constructs. For the *OsCIN1* promoter-driven GUS construct, we amplified a 2.0-kilobase promoter region of *OsCIN1* from the rice genome and inserted it into plasmid pCAMBIA1305, generating p*OsCIN1*::GUS. All plasmids were introduced into *Agrobacterium tumefaciens* strain LBA4404, and rice transformation was performed as described previously [41].

### 4.3. Quantitative Real-Time PCR

Total RNA was extracted from rice tissues using RiboEx reagent (GeneAll Biotechnology Co., Ltd., Seoul, Republic of Korea), and 1–5 µg of total RNA was used for cDNA synthesis using the ReverTra Ace qPCR RT Kit (Toyobo, Osaka, Japan). The cDNA was diluted 10-fold and stored at −20 °C. Quantitative RT-PCR was performed as described previously with *OsCIN1* primer (F: TGAGAAGCTTGATTGACCGTTC; R: ATAAGCGGCTTCTTCATTTCCC), using *OsUBQ5* (F: CCTCGCCGACTACAACATC; R GCTTGTGCTTCTGCTTCTTG) as the internal reference gene for normalization with triplicate [42].

### 4.4. CIN Activity

Leaf and stem base samples were harvested and immediately frozen in liquid nitrogen to preserve enzyme activity. For stem base samples, 1.5-cm sections were collected from the base of the stem. CIN activity was assayed using a method adapted and modified from Roitsch and González [43].

### 4.5. Subcellular Localization

The p35S::*OsCIN1*-GFP construct was transformed into rice callus to examine subcellular localization. Confocal laser scanning microscopy was performed using a Zeiss LSM 510 META confocal microscope (Carl Zeiss, Oberkochen, Germany). GFP fluorescence was excited using a 488-nm argon laser, and emission signals were collected at 500–550 nm. Images were processed and analyzed using ImageJ 1.53e software (National Institutes of Health, Bethesda, MD, USA).

### 4.6. Flag Leaf Metabolite Measurement

Fresh flag leaf samples were harvested and extracted using 10% perchloric acid (prepared from 37% stock solution). The insoluble and soluble fractions were separated by centrifugation at 13,000× *g* for 15 min. The soluble fraction (supernatant) was neutralized with neutralization buffer (2 M KOH, 0.4 M MES) to pH 6–7. The concentrations of glucose, fructose, and sucrose were determined through an NAD(P)H-coupled enzymatic assay by sequentially adding 0.5 units of hexokinase, 0.5 units of phosphoglucose isomerase, and 2 units of invertase. The measured metabolite contents were calculated relative to sample fresh weight. The insoluble fraction was washed repeatedly with 80% ethanol to remove chlorophyll until the pellet was colorless. The pellet was resuspended in an appropriate volume of water, and starch was gelatinized at 95 °C for 15 min. The gelatinized solution was incubated with 100 mM sodium acetate buffer (pH 4.8) containing 1 unit of amyloglucosidase and 1 unit of α-amylase at 37 °C overnight. After incubation, the mixture was centrifuged at 13,000× *g* for 5 min, and the glucose content in the supernatant was quantified using an NAD(P)H-coupled enzymatic assay to determine starch content [44].

### 4.7. GUS Staining

GUS staining of seeds and seedlings of the p*OsCIN1*::GUS line was conducted as described previously [45].

### 4.8. Field Trial

Field trials were performed at the LMO field of Gyeongsang National University in Sacheon, Republic of Korea. The paddy field was flooded with 5 cm of water until the end of the active tillering stage (approximately one month after transplanting). Water management was adjusted according to the growth stage, maintaining soil moisture during the late tillering stage, then re-flooding with 10 cm of water until the milky stage. To evaluate grain weight per plant, a total of eight plants of each line were grown in the paddy field. Seeds were harvested after ripening, dried, and weighed to determine grain weight per plant.

## 5. Conclusions

Our results highlight a critical principle for crop improvement: manipulating CIN activity to enhance yield demands precise spatiotemporal control rather than broad constitutive expression. Each *CIN* gene is likely specialized to reach the metabolic demands of specific tissues, developmental stages, and environmental conditions. The tissue-specific localization of *OsCIN1* in vascular tissues, particularly in ovular and lateral stylar vascular traces during seed development, demonstrates that it functions under specific spatial and temporal regulation. Future strategies to enhance sink capacity and improve crop yield must therefore be grounded in a comprehensive understanding of the spatial distribution of each CIN protein, the timing of their developmental stage–specific activation, and their functional integration with downstream hexose transporters.

## Figures and Tables

**Figure 1 ijms-26-11471-f001:**
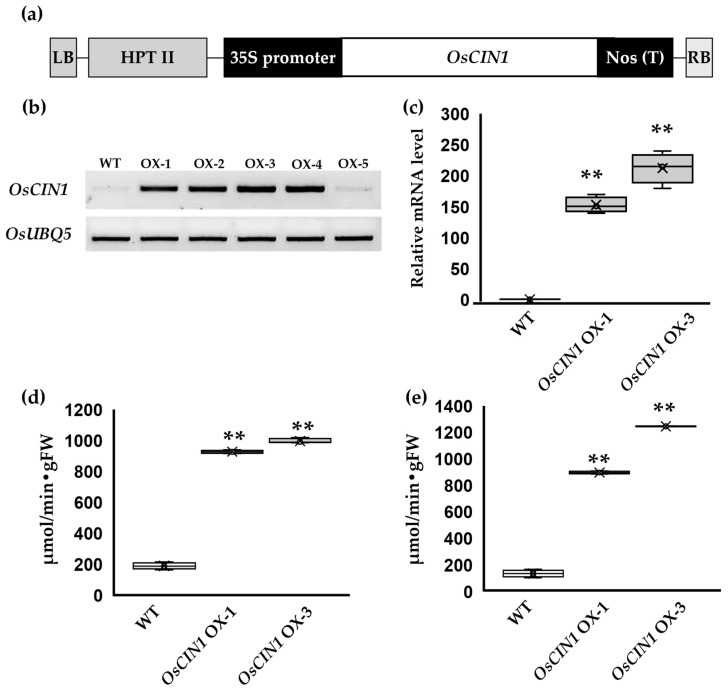
*OsCIN1* expression level and cell wall invertase activity in *OsCIN1* OX plants. (**a**) Schematic representation of the vector construct used to generate *OsCIN1* overexpressing rice plants. (**b**) RT-PCR analysis of *OsCIN1* expression in overexpressing plants. OsUBQ5 was amplified as an internal control. (**c**) qRT-PCR analysis of *OsCIN1* expression in overexpressing plants. OsACT1 was used as an internal reference gene. Relative expression levels are shown as fold change compared to WT. (**d**) Total cell wall invertase activity in fully expanded leaves of six-week-old plants. (**e**) Total cell wall invertase activity in stem base of plants at the flowering stage. Each value represents the mean ± SE of three biological replicates. Statistical analysis was performed using Student’s *t*-test. ** *p* < 0.01.

**Figure 2 ijms-26-11471-f002:**
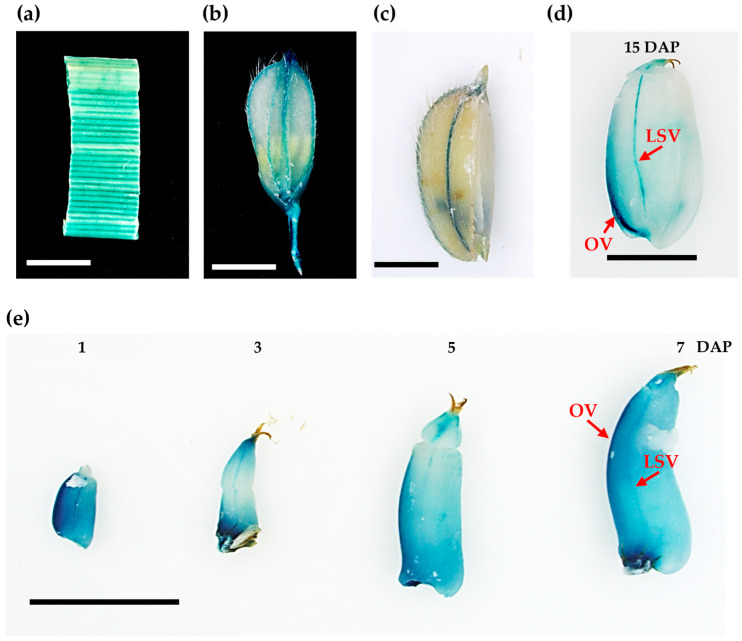
Tissue-specific expression pattern of p*OsCIN1*::GUS in rice. (**a**) GUS expression in the vascular tissues of a flag leaf of heading stage. (**b**) GUS expression in pre-fertilization young flower. (**c**) GUS expression in the lemma from the 15 DAP seed. (**d**,**e**) GUS expression during seed development at 1, 3, 5, 7, and 15 days after pollination (DAP). OV: ovular vascular traces; LSV: lateral stylar vascular traces. Scale bars = 5 mm.

**Figure 3 ijms-26-11471-f003:**
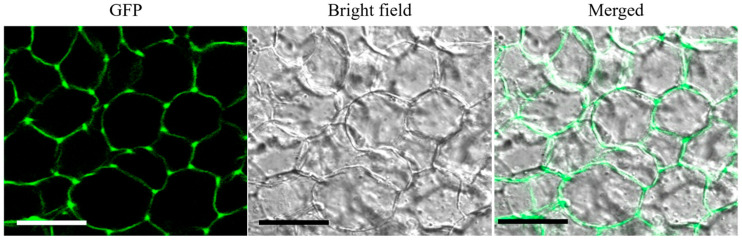
Subcellular localization of *OsCIN1* in the apoplast. Confocal microscopy images of *OsCIN1*-GFP fusion protein in transgenic rice calli. (**Left**) GFP fluorescence (green) showing *OsCIN1* localization to the cell wall and intercellular spaces. (**Middle**) Bright-field image. (**Right**) Merged image of GFP fluorescence and bright-field. The GFP signal indicates that *OsCIN1* localizes to the apoplast. Scale bar = 20 µm.

**Figure 4 ijms-26-11471-f004:**
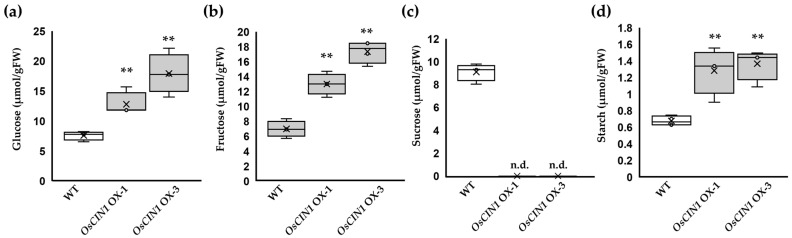
Soluble sugar and starch content in flag leaves of *OsCIN1* OX transgenic lines. Levels of soluble sugars: (**a**) glucose, (**b**) fructose, (**c**) sucrose, and (**d**) starch in flag leaves at the flowering stage. Two independent transgenic lines (*OsCIN1* OX-1 and *OsCIN1* OX-3) showed significantly altered sugar composition compared to WT plants. Each value represents the mean ± SE of three biological replicates. Statistical analysis was performed using Student’s *t*-test. ** *p* < 0.01. n.d., not detected.

**Figure 5 ijms-26-11471-f005:**
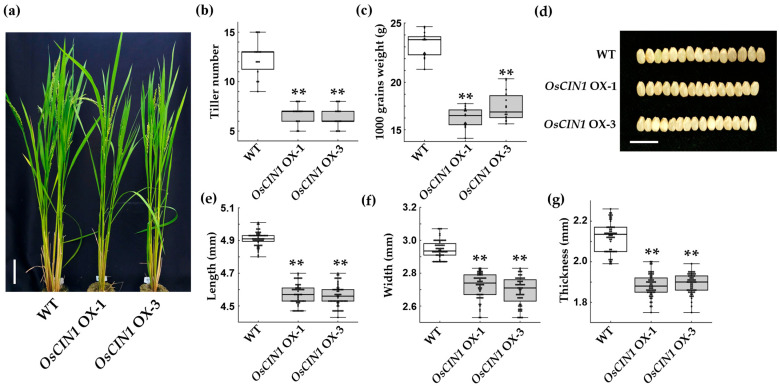
Phenotypic characterization of *OsCIN1* OX plants in paddy field conditions. (**a**) Representative plants of WT and OX lines at the flowering stage in the field. (**b**) Tiller number per plant. (**c**) 1000-grain weight. (**d**) Representative grain morphology showing width differences. (**e**–**g**) Grain length, width, and thickness, respectively. Scale bars, 10 cm (**a**); 1 cm (**d**). Each value represents the mean ± SE; n = eight plants (**b**) and n = 60 grains (**e**–**g**). Statistical analysis was performed using Student’s *t*-test. ** *p* < 0.01.

## Data Availability

No new data were created or analyzed in this study. Data sharing is not applicable to this article.

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
