# Peer review of "Constitutive OsCIN1 Expression Reprograms Source–Sink Dynamics and Compromises Agronomic Traits in Rice"

_ijms, 2025, doi:10.3390/ijms262311471_

Round 1

Reviewer 1 Report

Comments and Suggestions for Authors

In the above manuscript authors tried to establish the tissue specific and intracellular localization and the effect of ectopic overexpression on source-sink strength and overall growth and yield of OsCIN1 in rice. As the authors state in the discussion (lines 209-213) the objective was, “On this basis, we hypothesized that increasing OsCIN1 activity  would enhance the source-to-sink transport capacity of sink tissues and thereby improve crop productivity. To test this hypothesis, we generated OsCIN1 overexpression lines driven by the constitutive 35S promoter.”. The experimental approach is completely flawed and would not test the hypothesis. Ectopic expression of OsCIN1 with the CaMV 35S promoter in the whole plant constitutively in both source and sink tissues would not test this hypothesis. Rather you should have expressed OsCIN1 tissue specifically in the sink tissues such as developing seeds where enhanced sucrose hydrolysis can create a strong sink. Source-sink relationship requires a sucrose gradient for loading and then transport. If the sucrose levels are not detectable in the flag leaves due to hydrolysis of sucrose (Figure 4C), then the flag leaves cannot function as effective source tissues leading to yield reduction.  

               Authors state that the outcome of this experiment is unexpected under discussion in lines 218-22, “However, the phenotypic outcomes diverged from our initial expectations. Tiller number was markedly reduced in the overexpression lines, and 1,000-grain weight was also significantly decreased. Thus, constitutive overexpression of OsCIN1 throughout the plant did not improve productivity by strengthening sink capacity. Instead, it negatively affected important agronomic traits such as tillering and seed development”. The statement further indicates the authors did not thought about the hypothesis well. Nevertheless, the outcome was as expected provided the putative function of CIN1. Therefore, this a pure academic exercise rather than a well-planned approach based on a sound hypothesis.

However, the findings may be useful evidence as a proof of expected outcomes of the ectopic overexpression of OsCIN1.

  1. It is well established and accepted now that bar graphs do not represent the data very well. Just the mean and the SE is not enough. Show all the data points in all the bar graphs.
  2. Figure 2, pOsCIN1::GUS expression: Authors have performed the above experiment to show that pOsCIN1::GUS expresses mainly in vascular tissues during seed development and in leaf veins during early vegetative stages when young leaves may function as sinks. However, the results were presented very abstract and prevents us from seeing that OsCIN1 is not expressed in other tissues. See more details below.

(a) Authors report in lines 110-113 (also note the “comma” followed by a capital “S”), “GUS activity was observed predominantly in leaf veins (Fig. 2a) and in the vascular tissues of the lemma and palea (Fig. 2b), Strong expression was also observed in the ovular vascular trace and lateral stylar vascular traces during early grain-filling stages (Fig. 2e)”. Are both these observations observed in early grain filling stages or the former observation observed in young seedling? Depends on the answer, correct the sentence(s). Can I also see whole seedlings and whole tissues (e.g. a leaf rather than a tiny piece of a leaf) stained in the supplementary figures to verify that OCIN1 is indeed not expressed elsewhere during these developmental stages?

(b) Similar to (a) above I would like to see the other stained tissues (in Supplementary figures) to see the evidence for the statement in lines 113-115, “At 15 days after pollination (DAP), GUS activity was specifically localized to the ovular and lateral stylar vascular traces (Fig. 2d)”.

Comments on the Quality of English Language

Generally well written. Needs grammar check and sentence structure edits in places (copy editing)

Author Response

Response to Reviewer 1

Dear Reviewer,

We would like to express our gratitude to you for your valuable feedback. In response, we have made substantial revisions in line with your suggestions. Additionally, the manuscript has been proofread for English language accuracy, and major changes are highlighted in yellow.

Below, we provide our detailed responses to your comments.

  1. Authors state that the outcome of this experiment is unexpected under discussion in lines 218-22, “However, the phenotypic outcomes diverged from our initial expectations. Tiller number was markedly reduced in the overexpression lines, and 1,000-grain weight was also significantly decreased. Thus, constitutive overexpression of OsCIN1 throughout the plant did not improve productivity by strengthening sink capacity. Instead, it negatively affected important agronomic traits such as tillering and seed development”. The statement further indicates the authors did not thought about the hypothesis well. Nevertheless, the outcome was as expected provided the putative function of CIN1. Therefore, this a pure academic exercise rather than a well-planned approach based on a sound hypothesis.

Response: To address this comment, we have added the following paragraph to clarify the scientific rationale underlying our hypothesis and experimental design. “In addition, our recent functional genetic analysis demonstrated that the oscin1/oscin2 double mutant exhibits severe reductions in both seed starch accumulation and pollen starch synthesis. Enzyme activity assays further showed that OsCIN activity in mature anthers was approximately 500-fold higher than in leaves [22]. Taken together, these findings indicate that OsCIN1 and OsCIN2 act as the primary regulators of sink strength during reproductive development. This functional importance, combined with the role of CINs in enhancing crop productivity across multiple species [18,23, 27], led us to hypothesize that constitutive overexpression of OsCIN1 would enhance sink capacity in developing seeds by maintaining high apoplastic sucrose hydrolysis activity, thereby accelerating carbon flux into grains and improving crop yield. To test this hypothesis, we generated transgenic OsCIN1-OX lines under the CaMV 35S promoter.” We believe that this additional explanation strengthens and clarifies the justification for our experimental design and addresses the reviewer’s concern regarding the formulation of our original hypothesis.

  1. It is well established and accepted now that bar graphs do not represent the data very well. Just the mean and the SE is not enough. Show all the data points in all the bar graphs.

Response: Thank you for this important suggestion. We have replaced all bar graphs with box plots and now display all individual data points.

  1. Figure 2, pOsCIN1::GUS expression: Authors have performed the above experiment to show that pOsCIN1::GUS expresses mainly in vascular tissues during seed development and in leaf veins during early vegetative stages when young leaves may function as sinks. However, the results were presented very abstract and prevents us from seeing that OsCIN1 is not expressed in other tissues. See more details below. (a) Authors report in lines 110-113 (also note the “comma” followed by a capital “S”), “GUS activity was observed predominantly in leaf veins (Fig. 2a) and in the vascular tissues of the lemma and palea (Fig. 2b), Strong expression was also observed in the ovular vascular trace and lateral stylar vascular traces during early grain-filling stages (Fig. 2e)”. Are both these observations observed in early grain filling stages or the former observation observed in young seedling? Depends on the answer, correct the sentence(s). Can I also see whole seedlings and whole tissues (e.g. a leaf rather than a tiny piece of a leaf) stained in the supplementary figures to verify that OCIN1 is indeed not expressed elsewhere during these developmental stages? (b) Similar to (a) above I would like to see the other stained tissues (in Supplementary figures) to see the evidence for the statement in lines 113-115, “At 15 days after pollination (DAP), GUS activity was specifically localized to the ovular and lateral stylar vascular traces (Fig. 2d)

Response: Thank you for your detailed comments regarding the pOsCIN1::GUS expression analysis. To clarify the developmental stages at which each GUS-stained sample was collected, we have added the exact sampling time points directly to the figure panels. In addition, as requested, we have included GUS staining results from young seedlings in Supplementary Figure S1, allowing the reviewer to examine OsCIN1 expression across a broader range of developmental stages. Furthermore, in Figures 2d and 2e, we have marked the OV (ovular vascular traces) and LSV (lateral stylar vascular traces) with arrows for clearer visualization. The pOsCIN1::GUS expression analysis revealed that, across all developmental stages we examined, OsCIN1 expression is strongly enriched in the vascular tissues of leaves and the coleoptile, flowers, lateral roots, lateral root primordia, and developing seeds. We believe that this vascular-specific expression pattern provides important insights into the physiological function of OsCIN1 and also helps explain the phenotypes observed in the OsCIN1 overexpression lines.

Reviewer 2 Report

Comments and Suggestions for Authors

A review was performed on the manuscript entitled "
Constitutive OsCIN1 Expression Reprograms Source–Sink Dynamics and Compromises Agronomic Traits in Rice".
The revisions must be performed to address the core weaknesses of the original manuscript, significantly elevating its scientific rigor, clarity, and novelty. Authors should review the entire text and italicize genes. I suggest that authors look at the rules again and apply them to the manuscript. 
The manuscript would benefit from another round of review.

Author Response

Response to Reviewer 2

Dear Reviewer,

We would like to express our gratitude to you for your valuable feedback. In response, we have made substantial revisions in line with your suggestions. Additionally, the manuscript has been proofread for English language accuracy, and major changes are highlighted in yellow.

Below, we provide our detailed responses to your comments.

  1. The manuscript contains numerous grammatical and typographical errors, which hinder readability. Introduction needs better elaboration. Look for recent studies and remove those that are more than 5 years old unless they are important. The authors must add some novel contributions to the existing model to make it useful for the contributing society. Major revisions are necessary to improve the clarity, rigor, and scientific contribution of the study.

Response: We substantially revised the Introduction to improve clarity and strengthen the scientific context by incorporating recent studies from the past five years and removing older references unless essential. We also corrected grammatical and typographical errors throughout the manuscript. In addition, the Discussion section was reorganized to achieve better logical flow and coherence, thereby enhancing the overall rigor and scientific contribution of the study.

  1. In scientific writing, never use common names. Use scientific names instead. Correct this throughout your text.

Response: We have revised all names to their scientific forms.

  1. Never start a sentence with a number. Rewrite it.

Response: We have revised the sentences

  1. Please describe in more detail the methodology used. Were the analyses performed in duplicate or triplicate? Were the normalizers used based on any study? Please specify.

Response: We have revised the sentence to: “Quantitative RT-PCR was performed as described previously with OsCIN1 primer (F: TGAGAAGCTTGATTGACCGTTC; R: ATAAGCGGCTTCTTCATTTCCC), using OsUBQ5 (F: CCTCGCCGACTACAACATC; R: GCTTGTGCTTCTGCTTCTTG) as the internal reference gene for normalization with triplicate [42].” In addition, we have included the full primer sequences and added the corresponding reference as requested.

  1. Was the applied statistical analysis sufficient? Did it answer all the questions?

Response: Thank you for this important comment. We carefully re-evaluated the statistical analyses used in this study. For Figures 1 and 4, all quantitative data were obtained from independent biological replicates (n = 3, as indicated in the figure legends). For Figure 5, we have clearly specified the number of plants or seeds measured for each analysis. In response to the reviewer’s concerns, we have now explicitly stated in each figure legend the number of biological replicates, the statistical methods used, and the significance thresholds applied. In addition, all bar graphs have been replaced with box plots, and individual data points are now displayed. We believe that these statistical approaches are appropriate and sufficient to support the conclusions presented in this study.

Round 2

Reviewer 1 Report

Comments and Suggestions for Authors

None

Comments on the Quality of English Language

Generally well written. Needs grammar check and sentence structure edits in places (copy editing)